# 2021/22 and 2022/23 Post-Pandemic Bronchiolitis Seasons in Two Major Italian Cities: A Prospective Study

**DOI:** 10.3390/children10061081

**Published:** 2023-06-20

**Authors:** Anna Camporesi, Rosa Morello, Ugo Maria Pierucci, Francesco Proli, Ilaria Lazzareschi, Giulia Bersani, Piero Valentini, Damian Roland, Danilo Buonsenso

**Affiliations:** 1Department of Pediatric Anesthesia and Intensive Care, Children’s Hospital “Vittore Buzzi”, Via Ludovico Castelvetro 32, 20154 Milano, Italy; 2Department of Woman and Child Health and Public Health, Fondazione Policlinico Universitario A. Gemelli IRCCS, 00168 Rome, Italy; 3Department of Pediatric Surgery, Children’s Hospital “Vittore Buzzi”, 20154 Milano, Italy; 4Paediatric Emergency Medicine Leicester Academic (PEMLA) Group, Leicester Hospital, Leicester LE1 5WW, UK; 5Social Science APPlied to Healthcare Improvement Research, SAPPHIRE Group, Health Sciences, Leicester University, Leicester LE1 7RH, UK; 6Center for Global Health Research Studies, Università Cattolica del Sacro Cuore, 00168 Rome, Italy

**Keywords:** bronchiolitis, respiratory syncitial virus, COVID-19

## Abstract

**Objectives:** Bronchiolitis remains a major cause of morbidity and mortality in children under 24 months. During the first year of the pandemic, non-pharmacological interventions resulted in a significant reduction of bronchiolitis cases. Early in 2021, a rebound of bronchiolitis was reported with a description of out-of-season outbreaks. In this study, we prospectively evaluated the impact of bronchiolitis in two Italian University centers located in different geographical areas, aiming to compare two post-pandemic bronchiolitis seasons (2021/22 and 2022/23) in terms of severity, outcomes, microbiology and temporal distribution. **Methods:** This was a bicentric prospective observational cohort study. All consecutive children under 24 months of age assessed in the participating institutions during the specified seasons and receiving a clinical diagnosis of bronchiolitis were included. **Results:** A total of 900 patients were enrolled. Patients in the second season were globally younger and had comorbidities less often. Temporal distribution changed between the two seasons. Of the patients, 56% were tested for RSV; 60% of these was positive. Patients with RSV were globally younger (3.5 months vs. 4.9, *p* < 0.001), more often had a need for any kind of respiratory and fluid support and more often needed ward or PICU admission. At the end of the ED visit, 430 patients were discharged home, 372 (41.3%) were admitted to an inpatient ward and 46 (5.1%) to a pediatric intensive care unit. **Conclusions:** The 2022/23 post-COVID bronchiolitis was mostly similar to that of 2021/22, and was in line with pre-pandemic expectations.

## 1. Introduction

Bronchiolitis remains one of the major causes of morbidity and mortality in children younger than 24 months globally [1,2]. Although recent studies have shown promising results from passive immunization strategies to prevent severe disease in children during bronchiolitis seasons [3,4], their widespread implementation is not expected earlier than 2024 and, therefore, bronchiolitis is still expected to causes significant stress on the healthcare systems in future. 

During the first year of the pandemic, non-pharmacological interventions resulted in an unexpected impact on bronchiolitis seasons in every country, in terms of a significant reduction or even disappearance of new cases in the 2020/21 season [5]. The impact was particularly relevant on respiratory syncytial virus (RSV) circulation. However, early in 2021, the media and researchers reported a rebound of viral bronchiolitis with the description of early, out-of-season outbreaks [6]. Most authors hypothesized that a lack of exposure to the viruses during the previous year caused an immunity debt that led to a more susceptible population during the subsequent year [7]. Despite many claims on media and social media, many authors described the 2021/22 bronchiolitis seasons as having an overall severity comparable to pre-pandemic seasons, although apparently larger numbers of children were assessed in the emergency departments and during unusual periods of time [8,9,10,11,12]. A large nationwide study in Denmark including children up to five years of age compared post-pandemic seasons with four pre-pandemic seasons. It documented that, while overall bronchiolitis severity was similar, RSV had a more severe impact on older-than-usual children in the post-COVID era, hypothesizing that having a first RSV infection at an older age could be associated with more severe disease [13]. Hypothetically, as a large number of children had been exposed to a significant bronchiolitis season in 2021/22, it is expected that the last 2022/23 season is comparable to previous ones in terms of numbers of involved children and clinical severity. Therefore, it is pivotal to continue to prospectively monitor the burden of bronchiolitis in the post-pandemic world, in light of the future implementation of new pharmacological preventive interventions. In fact, it is expected that anti-RSV monoclonals and maternal vaccination will be implemented in 1–2 years, and to know their impact it is important to know the burden of post-pandemic bronchiolitis seasons.

In this study, we prospectively evaluated the impact of bronchiolitis in two University centers of two major Italian cities located in different geographical areas, aiming to compare two post-pandemic bronchiolitis seasons (2021/22 and 2022/23) in terms of disease severity, outcomes, microbiology and temporal distribution. The usual “RSV seasons” (which overlap with the bronchiolitis seasons) ranged in Italy from mid-November to the end of April, peaking in mid-February. In a previous paper from our group, we described the epidemiological and temporal modifications of the first post-lockdown bronchiolitis season. Here, we add data about the second post-lockdown season.

## 2. Methods

This is a multi-center observational cohort study performed in two Italian university hospitals in two different geographic areas (Northern Italy (Milano) and Central Italy (Rome)) from 1 July 2021 to 31 March 2022 (Season 1) and from 1 July 2022 to 31 March 2023 (Season 2). The study was adapted from an open access BronchSTART protocol developed by the PERUKI network, aiming to monitor real-time new bronchiolitis cases in the United Kingdom and Ireland since July 2021 [14]. The study was approved by the Institutional Review Boards of both centers. 

All consecutive children under 24 months of age assessed in the participating institutions and receiving a clinical diagnosis of bronchiolitis were included in the study. Bronchiolitis was defined as the presence of respiratory distress and signs of lower respiratory tract infection [1] (e.g., cough, tachypnoea or chest recession, and wheeze or crackles on chest auscultation) or a first episode of acute viral wheeze. The upper limit of age was intentionally set at 24 months because there is no full agreement on the upper age limit for diagnosing bronchiolitis according to different international guidelines [1], and because children who were in their second year of life in the first season might have missed being in contact with common respiratory viruses linked to bronchiolitis due to restrictive measures during their first year of life. The hypothesis was that this age group might have suffered a first episode of bronchiolitis or viral wheezing. 

To achieve microbiological diagnosis, children were tested according to the decision of the evaluating physician. All children in both centers received rapid antigenic SARS-CoV-2 testing, as per national anti-COVID policies. For the other respiratory viruses, we used the Allplex™ Respiratory Panel 1, 2, 3 Seegene System, which is a molecular method for the genome detection and typing of the following respiratory viruses: respiratory syncytial virus (RSV A and B), influenza virus (flu A and B), parainfluenza viruses (PIV 1, 2, 3 and 4), adenovirus (AdV), enterovirus (HEV), metapneumovirus (MPV), rhinovirus (HRV), bocavirus (HBoV) and coronavirus (CoV NL63, 229E, OC43). Tests were conducted on nasal swabs.

The primary aim of the study is to describe the epidemiology of bronchiolitis in the first two seasons after the beginning of the COVID-19 pandemic and relative lockdowns in two different Italian regions and describe potential differences between the two seasons. In this paper, we added the second post-lockdown bronchiolitis season to the analysis in order to compare differences between them. Secondary outcomes are the determination of:
-The percentage of cases due to RSV in both seasons and relative differences; -The percentage of cases due to SARS-CoV-2 in both seasons and relative differences; -Differences in disease severity and support needed in RSV vs. non-RSV cases and COVID-19 vs. non-COVID-19 cases;-Risk factors for need of HFNC (high flow nasal cannula) and PICU (pediatric intensive care unit) admission.

## 3. Statistical Methods

Data were analyzed with Stata 17.0 BE (StataCorp LLC, USA). Categorical variables were described as frequencies and percentages, and continuous variables expressed as the mean (±standard deviation) [median; interquartile ranges]. We used the chi-square test to analyze categorical variables and Student’s *t*-test for continuous variables or the Wilcoxon rank-sum test as appropriate. Statistical significance was designated as a *p* value ˂ 0.05 (2-sided) or a 95% confidence interval that does not include 1.

The occurrence of PICU admission and need for HFNC were considered as the dependent variables in two respective logistic regression models. Univariate variable selection (likelihood ratio test) using *p* < 0.25 to select candidates for the multivariable model was performed on the following possible predictive variables: age (months), ex-prematurity status (defined as birth before 37 gestational weeks), newborn status (defined as age under 28 days), toddler status (defined as age > 12 months) and the presence of known bronchopulmonary dysplasia, congenital cardiac disease, neuromuscular disorder or other comorbidities; vital parameters upon admission were: SpO2, respiratory rate, nasal aspirate positive for RSV or SARS-CoV-2, pH and pCO_2_ on admission. Goodness-of-fit of the models was assessed with Hosmer–Lemeshow test. 

Similarly, a multivariate linear regression model was built taking SpO2 upon admission as the dependent variable. Variables included in the final model were checked for multicollinearity by the variance inflation factor (VIF).

## 4. Results

During the study period, 900 patients were enrolled in the study in the two seasons in the two centers. 

Milano enrolled 163 patients in the first season and 309 in the second season; Rome enrolled 178 patients in the first season and 250 in the second one. 

Demographic characteristics of the patients are reported in Table 1.

Patients in the second season (2022–23) were globally younger and less often had a diagnosis of bronchopulmonary dysplasia or neuromuscular disorder and the presence of other comorbidities. 

Data about siblings of patients are available for 329 patients, who had one or more. 

There were 222 newborns (24.7% of the population). Ex-premature babies represent 9.9% of the population (n = 89). A total of 11 patients in the second season and 2 in the first season received palivizumab prophylaxis.

### Temporal Distribution of Cases in the Two Seasons

The overall temporal distribution of the cases is represented in Figure 1. During the 2021–22 season, the first cases were noted in the middle of the Italian summer (July), an anomalous report at our latitudes; they then peaked in November and December and showed a fast decline during the second half of December. During the 2022–23 season, the first cases appeared in October, rose significantly in November and December, peaked at the end of December and beginning of January, then declined.

## 5. Microbiology

A total of 503 (56%) patients received point-of-care virus testing for RSV: 189 in the first season and 314 in the second one. Of these, 59% (295) was positive for RSV: 105 on 189 tests executed in the first season and 202/314 in the second season. 

All patients received a point-of-care COVID-19 test during both seasons in both centers. Of these, 33 patients were positive for COVID-19 (28 in the first season and 5 in the second season) (Figure 2). Having COVID-19 as an etiological agent of bronchiolitis showed a strong association with the season (*p* < 0.001,) while having RSV as an etiological agent showed no correlation (*p* = 0.103). Children in the second season were significantly younger (*p* = 0.0094) and without comorbidities (*p* = 0.001).

### 5.1. Severity Associated with Virus

Patients with RSV were globally younger (3.5 months vs. 4.9, *p* < 0.001), more often had a need for suction, any kind of respiratory support and fluid support and more often needed ward or PICU admission (Table 2 and Table 3).

### 5.2. Peripheral Saturation of Oxygen (SpO2) and Respiratory Rate

SpO2 data are available for 873 patients out of 900. Mean SpO2 was 95 (±4) [97; 93–98]% and did not differ significantly between the two seasons. 

Mean SpO2 in patients admitted to ward was 94 (±5) [94; 90–98]%. Mean SpO2 in patients admitted to PICU was 87 (±7) [88.5; 85–90]%. RR was 53 (±11) [52; 44–60] with no significant differences between seasons.

Univariate and multivariable linear regression was conducted for the outcome “SpO2”.

In the multivariable analysis, significant determinants were age (months) (Coeff: −0.24, 95% CI: −0.32–−0.16, *p* < 0.001), newborn status (Coeff: −1.46; 95% CI: −2.22–−0.61, *p* = 0.001), ex-premature status (Coeff: −1.1, 95% CI: −2.11–−0.09, *p* = 0.033), presence of congenital cardiopathy (Coeff: −2.26, 95% CI: −4.49–−0.03, *p* = 0.046) and detection of RSV (Coeff: −1.75; 95% CI: −2.40–−1.10; *p* < 0.001) in nasal aspirate. 

### 5.3. Blood Gas Analysis (BGA)

Overall, 126 patients received a blood gas analysis during the ED visit. Of these, 65 (52%) were arterial BGA and the remaining venous. Mean arterial pH was 7.36 (±0.06) [7.38; 7.34–7.4]; mean venous pH was 7.35 (±0.07) [7.35; 7.28–7.39]. Mean arterial CO_2_ was 42.1 (±9.9) [40; 34.8–49.6)] mmHg and mean venous CO_2_ was 47.9 (±12.3) [47; 40–52] mmHg. There was no significant difference between mean pH and CO_2_ values between the two seasons.

### 5.4. Support

Across both seasons, 42.1% of patients needed no support therapy while in the ED with an increase from the first to the second season. In the more recent season, more patients received intravenous fluids and less patients were treated with CPAP while in the ED. Use of nasogastric fluids, low-flow oxygen, HFNC and intubation did not change between seasons (Table 4). 

Two patients (0.02%) were intubated while in the ED, one for each season. 

### 5.5. Ex-Premature Status

The need for support in this sub-population was then analyzed according to their gestational age (extremely premature: ≤28 weeks; very premature: 29–32 weeks; moderately premature: 33–37 weeks). Need for fluids and low flow oxygen was more frequent in the extremely premature ones. Being an ex-premature was globally significantly associated with admission to PICU (*p* = 0.024) and use of HFNC (*p* < 0.001). There was no significant difference between seasons. 

### 5.6. Outcomes

Overall, at the end of the ED visit, 430 patients were discharged home, 49 (5.4%) were admitted to a short observation unit, 372 (41.3%) were admitted to an inpatient ward, 9 (1%) to a high-dependency unit and 46 (5.1%) to a pediatric intensive care unit. No patient died in the ED. From the first season to the second one, there was an increase in admission to short stay units and high-dependency units (Table 5). 

Univariate and multivariable linear regression was conducted for the outcomes “PICU admission” and “HFNC ventilation”. 

In multivariate analysis, the single most relevant factor correlating with PICU admission was RSV positivity (OR: 2.84, 95% CI: 1.54–5.33, *p* = 0.001). For the outcome “HFNC” use, significant factors were RSV positivity (OR: 2.43, 95% CI: 1.46–4.03, *p* = 0.001) and prematurity (OR: 3.84, 95% CI: 2.08–7.07, *p* < 0.001). 

## 6. Discussion

In this study, we compared two post-pandemic bronchiolitis seasons in two major Italian cities in two different geographical areas of the country. Overall, we found that the two seasons were similar in terms of clinical outcomes (e.g., PICU admission, respiratory support). The second season was characterized by a slightly larger number of patients evaluated during a shorter and more characteristic period of the year for a bronchiolitis season. Interestingly, significantly less children had SARS-CoV-2 as an etiological agent of bronchiolitis and children during the second season were significantly younger and without comorbidities. These data will be particularly useful to plan future strategies to implement pharmacological preventive interventions such as maternal vaccination and long acting monoclonals.

It is relevant to note that the large majority of children were younger than six months and without comorbidities, and therefore not eligible for current RSV prophylaxes. Of note, 60% of the tested children had a positive result for RSV, which may have several implications. As noted, RSV-positive children have a significantly higher risk of hospitalization, including PICU admission, despite less frequently having comorbidities such as prematurity. As their mean age was 3.5 (±3.8) months, the large majority of severe RSV cases in Italy would have been prevented by maternal RSV vaccination, according to the recent studies showing its major effectiveness in the first 3–5 months [3]. However, the older ones may only be partially protected and, since the temporal distribution of the last season was classic (lasting around 5–6 months), most children would also benefit from the new long-acting monoclonals [4]. Although the final decision on which intervention to prioritize is mostly based on the capability of each system to implement one strategy over the other and on cost-effective benefits, it is still important to highlight that at least one third of children assessed in hospitals with bronchiolitis are not RSV positive, and therefore would not benefit from any intervention. Although we found that less children with non-RSV bronchiolitis are hospitalized in a regular ward or PICU, still two hundred children in the two hospitals remained in observation for at least two days. This means that although it may be predictable that the burden on health systems of bronchiolitis will change in future with vaccines and monoclonals, bronchiolitis will remain an important cause of hospitalization and hospitals need to remain alert in terms of preparedness and response. 

In a sub-analysis of disease severity according to microbiology, we found that RSV bronchiolitis is more severe than other non-RSV bronchiolitis, even when compared with SARS-CoV-2 positive cases. Interestingly, only a minority of cases of bronchiolitis were due to SARS-CoV-2, and even less cases have been reported in the second season, despite mandatory SARS-CoV-2 testing in all children with respiratory symptoms still being in place in our institution. This is in line with previous studies showing that SARS-CoV-2 is neither frequent nor a more severe cause of bronchiolitis in children [8,15]. The reasons why SARS-CoV-2 is a very rare cause of bronchiolitis are unclear. Last year we hypothesized that a mechanism of viral interference might play a role, with other common respiratory viruses stimulating upper airway responses that prevent SARS-CoV-2 replication, as found with other viruses such as influenza and rhinovirus [16,17]. Another hypothesis is that maternal immunity to SARS-CoV-2 (either by previous infection or vaccination, or both) may protect the large majority of infants from COVID-19 bronchiolitis. This scenario is reasonable considering that the large majority of women have had multiple vaccinations in Italy, given the strong vaccination campaign implemented during the first years of the pandemic, and many have also been reinfected since the beginning of the Omicron waves. There is now plenty of evidence that anti-SARS-CoV-2 IgGs cross the placenta and are also found in the breastmilk, therefore protecting the babies [18,19,20,21]. 

In terms of outcomes, the two seasons were mostly similar. The same proportion of children required ward or PICU admission, suggesting that there are no factors that, during this pandemic, led to a higher risk of severe disease in children with bronchiolitis. Some authors have speculated that the lack of circulation of common viruses in the general population may have impaired the normal development of the immune system in children, while others have suggested that previous SARS-CoV-2 infection may lead to transient immune dysfunction, therefore favoring possible worse outcome in children infected with other pathogens shortly after [22,23]. Moreover, multivariate analyses found RSV positivity as a significant risk factor for PICU admission and high-flow oxygen support, as expected, and prematurity remained a significant risk factor for severe outcomes in both seasons, as expected. Therefore, our findings, for the second year in a row, do not reflect a changed overall severity or significantly higher number of children assessed in the two periods. However, an interesting finding was that, during the second season, the mean age of the children evaluated for bronchiolitis was significantly lower. Although no formal publications from the 2022/23 season have been reported already, on social media many colleagues have shared informal data about seeing a large number of younger-than-usual children with bronchiolitis, as was also found in our cohort. One of the reasonable hypotheses is that, during the pandemic, less young people, including women of childbearing age, had contact with RSV. As a consequence, a lower proportion of pregnant women may have had anti-RSV IgGs and, therefore, more young infants may have been less protected by maternal antibodies. This theory has a solid basis according to recent publications showing that pregnant women during the pandemic have lower rates of anti-RSV IgGs [22,23]. 

The major limitation of this study is that we measured outcomes within 24 h since the first evaluation. This means that if children worsened more than 24 h after initial admission and required transfer to PICU, that outcome was missed. In addition, not all children were systematically tested; therefore, the etiology was not noted for all children, since usually more severe cases were tested. For SARS-CoV-2, conversely, all children received at least a rapid antigenic test. 

In conclusion, our study shows that the 2022/23 post-COVID bronchiolitis was mostly similar to that of 2021/22, and overall was in line with pre-pandemic expectations. These data provide pivotal information for policymakers to plan the implementation of future pharmacological preventive strategies and, also, a post-pandemic baseline overview to measure the impact of these interventions.

## 7. Statements

The authors have no financial or non-financial interests that are directly or indirectly related to the work submitted for publication. 

The authors declare this work has not been published before and that it is not under consideration for publication anywhere else, and that its publication has been approved by all co-authors.

## Figures and Tables

**Figure 1 children-10-01081-f001:**
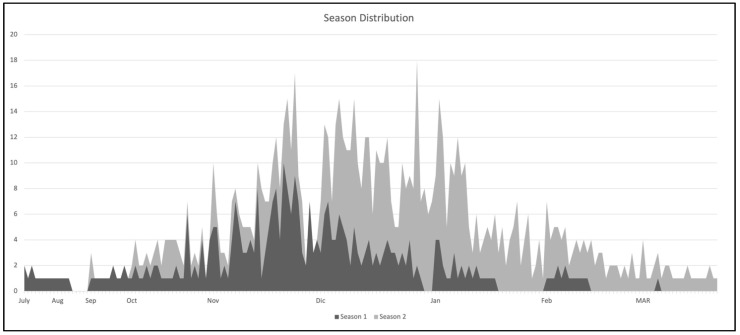
Temporal distribution of bronchiolitis admissions to the EDs in the participating centers in the two seasons considered.

**Figure 2 children-10-01081-f002:**
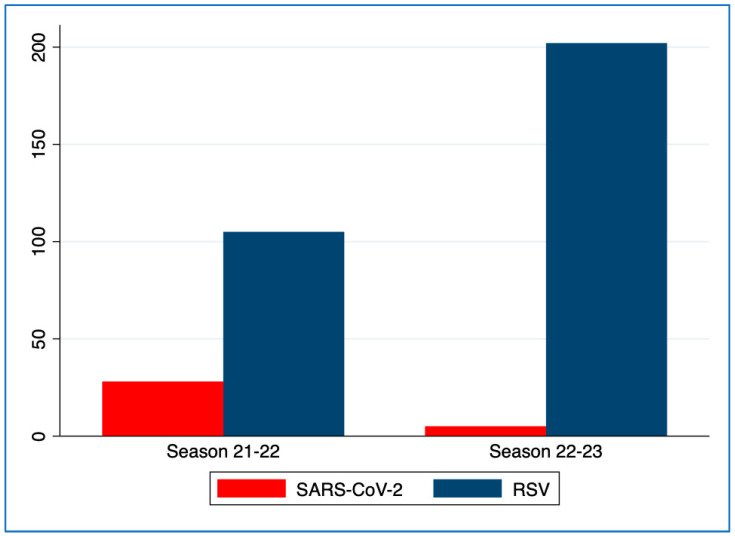
Graphical distribution of detected virus in nasal aspirates of patients admitted to the centers in the two seasons with diagnosis of bronchiolitis. RSV: Respiratory syncytial virus.

**Table 1 children-10-01081-t001:** Demographics of the study population. Data are presented as n (%). Newborns: age under 28 days. Toddlers: age 12–24 months.

	Total	Season 21–22	Season 22–23	*p*-Value
	N = 900	N = 341	N = 559	
**Age (months)**	4.4 (±4.4) [3.0; 1.0–6.0]	5.2 (±5.2) [3.0; 2.0–7.0]	4.0 (±3.7) [3.0; 1.0–5.0]	0.009
**Male sex**	528 (58.7%)	194 (56.9%)	334 (59.7%)	0.40
**Ex-premature status**	89 (9.9%)	28 (8.2%)	61 (10.9%)	0.19
**Bronchopulmonary dysplasia**	8 (0.9%)	6 (1.8%)	2 (0.4%)	0.030
**Congenital cardiac disease**	16 (1.8%)	7 (2.1%)	9 (1.6%)	0.63
**Neuromuscular disorder**	9 (1.0%)	6 (1.8%)	3 (0.5%)	0.074
**Other comorbidities**	61 (6.8%)	34 (10.0%)	27 (4.8%)	0.003
**Newborns**	222 (24.7%)	213 (24.6%)	9 (27.3%)	0.72
**Toddlers**	89 (9.9%)	86 (9.9%)	3 (9.1%)	0.88

**Table 2 children-10-01081-t002:** Epidemiological and clinical differences in presentation and need for support in patients infected with RSV and not.

	Total	No RSV	RSV	*p*-Value
	N = 898	N = 591	N = 307	
**Age (months)**	4.4 (±4.4) 3.0 [(1.0–6.0)]	4.9 (±4.5) 3.0 [(2.0–7.0)]	3.5 (±3.8) 2.0 [(1.0–4.0)]	<0.001
**Previous enrollment**	43 (4.8%)	35 (5.9%)	8 (2.6%)	0.027
**Male sex**	526 (58.6%)	366 (61.9%)	160 (52.1%)	0.005
**Ex-premature status**	88 (9.8%)	61 (10.3%)	27 (8.8%)	0.47
**Bronchopulmonary dysplasia**	8 (0.9%)	6 (1.0%)	2 (0.7%)	0.58
**Congenital cardiac disease**	16 (1.8%)	8 (1.4%)	8 (2.6%)	0.18
**Neuromuscular disorders**	9 (1.0%)	5 (0.8%)	4 (1.3%)	0.51
**Other comorbidities**	61 (6.8%)	37 (6.3%)	24 (7.8%)	0.38
**No comorbidities**	768 (85.5%)	509 (86.1%)	259 (84.4%)	0.48
**SpO2**	95.3 (±4.7) [97.0; 93.5–99.0]	95.9 (±4.5) [97.0;95.0–99.0]	94.3 (±4.8) [95.0; 91.0–98.0]	<0.001
**pH**	7.3 (±0.1) [7.3; 7.3–7.4]	7.3 (±0.1) [7.3; 7.3–7.4]	7.3 (±0.1) [7.3; 7.3–7.4]	0.37
**CO_2_**	45.5 (±11.6) [44.0; 37.0–52.0]	45.6 (±10.6) [42.6; 40.0–52.0)]	45.5 (±12.5) [45.3; 36.4–52.0]	0.91
**Therapy/Support**				
**Suction**	14 (1.6%)	3 (0.5%)	11 (3.6%)	<0.001
**No therapy**	378 (42.1%)	326 (55.2%)	52 (16.9%)	<0.001
**Nasogastric fluids**	1 (0.1%)	0 (0.0%)	1 (0.3%)	0.17
**Intravenous fluids**	116 (13.0%)	54 (9.2%)	62 (20.2%)	<0.001
**Oxygen (low flow)**	149 (16.6%)	65 (11.0%)	84 (27.4%)	<0.001
**HFNC**	68 (7.6%)	32 (5.4%)	36 (11.7%)	<0.001
**CPAP**	7 (0.8%)	2 (0.3%)	5 (1.6%)	0.038
**IMV**	2 (0.2%)	1 (0.2%)	1 (0.3%)	0.64
**Destination after ED**				
**Home discharge**	430 (47.9%)	372 (62.9%)	58 (19.0%)	<0.001
**Short stay unit**	49 (5.5%)	27 (4.6%)	22 (7.2%)	0.10
**Ward**	371 (41.3%)	167 (28.3%)	204 (66.4%)	<0.001
**HDU**	9 (1.0%)	4 (0.7%)	5 (1.6%)	0.17
**PICU**	45 (5.0%)	19 (3.2%)	26 (8.5%)	<0.001

SpO2: Peripheral oxygen saturation; HFNC: high flow nasal cannula; CPAP: continuous positive airway pressure: HDU: high dependence unit; PICU: pediatric intensive care unit.

**Table 3 children-10-01081-t003:** Epidemiological and clinical differences in presentation and need for support in patients infected with SARS-CoV-2 and not.

	Total	No COVID-19	COVID-19	*p*-Value
	N = 900	N = 867	N = 33	
**Age (months)**	4.4 (±4.4) [3.0; 1.0–6.0]	4.5 (±4.4) [3.0; 2.0–6.0]	3.5 (±3.0) [3.0; 1.0–4.0]	0.20
**Previous enrollment**	43 (4.8%)	40 (4.6%)	3 (9.1%)	0.24
**Male sex**	528 (58.7%)	509 (58.7%)	19 (57.6%)	0.90
**Ex-premature status**	89 (9.9%)	84 (9.7%)	5 (15.2%)	0.30
**Bronchopulmonary dysplasia**	8 (0.9%)	7 (0.8%)	1 (3.0%)	0.18
**Congenital cardiac disease**	16 (1.8%)	16 (1.8%)	0 (0.0%)	0.43
**Neuromuscular disorder**	9 (1.0%)	9 (1.0%)	0 (0.0%)	0.56
**Other comorbidities**	61 (6.8%)	57 (6.6%)	4 (12.1%)	0.21
**No comorbidities**	769 (85.4%)	739 (85.2%)	30 (90.9%)	0.36
**SpO2**	95.3 (±4.7) [97.0; 93.0–99.0]	95.3 (±4.7) [97.0; 93.0–98.0]	97.1 (±3.3) [98.0; 96.0–99.0]	0.005
**pH**	7.3 (±0.1) [7.3; 7.3–7.4]	7.3 (±0.1) [7.3; 7.3–7.4]	7.3 (±0.1) [7.3; 7.2–7.3]	0.18
**CO_2_**	45.5 (±11.6) [44.0; 37.0–52.0]	45.5 (±11.7) [44.0; 37.0–52.0]	52.0 (.) [52.0; 52.0–52.0]	0.37
**Therapy/Support**				
**No therapy**	378 (42.1%)	371 (42.9%)	7 (21.2%)	0.013
**Suction**	14 (1.6%)	14 (1.6%)	0 (0.0%)	0.46
**Nasogastric fluids**	1 (0.1%)	1 (0.1%)	0 (0.0%)	0.84
**Intravenous fluids**	116 (12.9%)	116 (13.4%)	0 (0.0%)	0.024
**Oxygen (low flow)**	150 (16.7%)	147 (17.0%)	3 (9.1%)	0.23
**HFNC**	68 (7.6%)	65 (7.5%)	3 (9.1%)	0.74
**CPAP**	7 (0.8%)	7 (0.8%)	0 (0.0%)	0.60
**IMV**	2 (0.2%)	2 (0.2%)	0 (0.0%)	0.78
**Destination after ED**				
**Discharge**	430 (47.8%)	411 (47.5%)	19 (57.6%)	0.25
**Short stay unit**	49 (5.4%)	47 (5.4%)	2 (6.1%)	0.87
**Ward**	372 (41.3%)	361 (41.6%)	11 (33.3%)	0.34
**HDU**	9 (1.0%)	9 (1.0%)	0 (0.0%)	0.56
**PICU**	46 (5.1%)	45 (5.2%)	1 (3.0%)	0.58

SpO2: Peripheral oxygen saturation; HFNC: high flow nasal cannula; CPAP: continuous positive airway pressure: HDU: high dependence unit; PICU: pediatric intensive care unit.

**Table 4 children-10-01081-t004:** Need for support in the population studied, according to season. Data are presented as n (%).

	Total	Season 21–22	Season 22–23	*p*-Value
	N = 900	N = 341	N = 559	
**No therapy**	378 (42.1%)	120 (35.3%)	258 (46.2%)	0.001
**Suction**	14 (1.6%)	6 (1.8%)	8 (1.4%)	0.69
**Feed/fluids**	254 (28.3%)	81 (23.8%)	173 (31.0%)	0.019
**Nasogastric fluids**	1 (0.1%)	1 (0.3%)	0 (0.0%)	0.19
**Intravenous fluids**	116 (12.9%)	20 (5.9%)	96 (17.2%)	<0.001
**Oxygen (low flow)**	150 (16.7%)	55 (16.2%)	95 (17.0%)	0.76
**HFNC**	68 (7.6%)	22 (6.5%)	46 (8.2%)	0.34
**CPAP**	7 (0.8%)	7 (2.1%)	0 (0.0%)	<0.001
**IMV**	2 (0.2%)	1 (0.3%)	1 (0.2%)	0.72

HFNC: High flow nasal cannula; CPAP: continuous positive airway pressure; IMV: invasive mechanical ventilation.

**Table 5 children-10-01081-t005:** Admission/discharge rates of infants presented to the EDs of the two centers during the given seasons. Data are presented as n (%).

	Total	Season 21–22	Season 22–23	*p*-Value
	N = 900	N = 341	N = 559	
**Discharged**	430 (47.8%)	172 (50.6%)	258 (46.2%)	0.20
**Ward**	372 (41.3%)	133 (39.0%)	239 (42.8%)	0.27
**Short Stay Unit**	49 (5.4%)	30 (8.8%)	19 (3.4%)	<0.001
**HDU**	9 (1.0%)	0 (0.0%)	9 (1.6%)	0.019
**PICU**	46 (5.1%)	17 (5.0%)	29 (5.2%)	0.90
**Death**	0 (0.0%)	0 (0.0%)	0 (0.0%)	

HDU: High-dependance unit. PICU: Pediatric intensive care unit.

## Data Availability

Data are available upon request to the corresponding author.

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
