# Peer review of "2021/22 and 2022/23 Post-Pandemic Bronchiolitis Seasons in Two Major Italian Cities: A Prospective Study"

_children, 2023, doi:10.3390/children10061081_

Round 1
Reviewer 1 Report
This is a very interesting study on the changing epidemiology of RSV bronchiolitis in post-Covid era. The topic is very interesting and definitely broadens our knowledge of this common disease of infants and very young children. There are to minor issues that need to be addressed:
1. Why were not all the children with bronchiolitis tested for RSV? What were the criteria for testing? Tose who were more sick? This could have biased the results
2. I am not sure that your results are an evidence agaist the "immunological debt" caused by Covid pandemics. In the 2 postpandemc seasons more children actually got sick according to European Centers for Disease Control than before pandemics. This study does not compare infection epidemiology before and after pandemicsso this conclusion is in no way supported by these data.
Author Response
Reviewer Response
Dear Reviewer,
thank you so much for your support. We have provided a point-by-point response to your comments. Changes have been highlighted in the main version of the manuscript
Reviewer 1
This is a very interesting study on the changing epidemiology of RSV bronchiolitis in post-Covid era. The topic is very interesting and definitely broadens our knowledge of this common disease of infants and very young children.
Thank you for your appreciation
There are to minor issues that need to be addressed:
- Why were not all the children with bronchiolitis tested for RSV? What were the criteria for testing? Tose who were more sick? This could have biased the results
Thank you for your comment. More severe cases were usually tested, as the protocol was observational for the diagnosis of bronchiolitis, not for the aetiology. Although it is possible this have biased results, we think it is a good representation as more than half of children were tested. We included this in the limitation section.
- I am not sure that your results are an evidence agaist the "immunological debt" caused by Covid pandemics. In the 2 postpandemc seasons more children actually got sick according to European Centers for Disease Control than before pandemics. This study does not compare infection epidemiology before and after pandemicsso this conclusion is in no way supported by these data.
You are right, we have delated that conclusion sentence as we have not compared formally pre-pandemic data. Just to note (this is not included in the paper) we have anyway had similar severities as expected for bronchiolitis, the more severe impact has been noted mostly in older children (these data are under review now)
Reviewer 2 Report
Major comments:
Introduction:
1)... Hypothetically, as a large number of children has been exposed to a significant bronchiolitis season in 2021/22, it is expected that the last 2022/23 season is comparable to previous ones. Therefore, this means its pivotal for continuous prospective monitoring of bronchiolitis in the post pan-demic world while considering the future implementation of new pharmacological pre-ventive interventions...The meaning of the sentences is not clear. Need to be rephrased.
Methods:
1) The last paragraph strating with... Primary aim of the study is to describe... in this section should be part of the introduction, not the method.
2) As stated in the paper that the usual "RSV seasons" in Italy range from mid-November to the end of April, explain why monitoring in both seasons was omitted from April to June, since by not including that period we lost a significant part of the RSV season
3) Virus detection methods are not described...see under comment for results.
Results:
The patient did not received but they were tested using pont of care tests. The tests used should be declared and briefly described (antigen or molecular) in the method section. Whether the same tests were used in both centers? Its sensitivity and specificity? What was the clinical sample (swab, aspirate or other)?
The following sentence should be deleted and only the parameters that showed a significant difference should be listed....“Although not always statistically significant, patients infected with SARS-CoV-2 showed a trend toward a higher incidence of ex-premature status, a higher SpO2, lower need for Oxygen support, and a tendency for lower admission rates to ward and PICU.“
Discussion:
„Interestingly, significantly less children had SARS-CoV-2 as an aetiologi-cal agent of bronchiolitis and children during the second season were significantly younger and without comorbidities.“ Please add the corresponding p values.
Minor comments:
1) The number at the beginning of the sentence should be written as a word (i.e. nine hundred, eleven, thirty three etc...); numbers up to ten are also written as a word (i.e. two).
2) Accompanying the tables, abbreviations must be explained as well as when they appear for the first time in the text
Author Response
Reviewer Response
Dear Reviewer,
thank you so much for your support. We have provided a point-by-point response to your comments. Changes have been highlighted in the main version of the manuscript.
Major comments:
Introduction:
1)... Hypothetically, as a large number of children has been exposed to a significant bronchiolitis season in 2021/22, it is expected that the last 2022/23 season is comparable to previous ones. Therefore, this means its pivotal for continuous prospective monitoring of bronchiolitis in the post pan-demic world while considering the future implementation of new pharmacological pre-ventive interventions...The meaning of the sentences is not clear. Need to be rephrased.
We have rephrased as follow: “Hypothetically, as a large number of children has been exposed to a significant bronchiolitis season in 2021/22, it is expected that the last 2022/23 season is comparable to previous ones in terms of numbers of involved children and clinical severity. Therefore, this means its pivotal to continue to prospectively monitor the burden of bronchiolitis in the post pandemic world, in light of the future implementation of new pharmacological preventive interventions. In fact, it is expected that anti-RSV monoclonals and maternal vaccination will be implemented in 1-2 years, and to know their impact it is important to know the burden of post-pandemic bronchiolitis seasons.”
Methods:
1) The last paragraph strating with... Primary aim of the study is to describe... in this section should be part of the introduction, not the method.
Corrected, thank you
2) As stated in the paper that the usual "RSV seasons" in Italy range from mid-November to the end of April, explain why monitoring in both seasons was omitted from April to June, since by not including that period we lost a significant part of the RSV season
It is true, although in April we have very limited number of cases, if any. However, as the period analyzed is the same in the two seasons, this should not have affected the overall significance of our findings. We have included this in the study limitation
3) Virus detection methods are not described...see under comment for results.
Ok, please find response under the other comment
Results:
The patient did not received but they were tested using pont of care tests. The tests used should be declared and briefly described (antigen or molecular) in the method section. Whether the same tests were used in both centers? Its sensitivity and specificity? What was the clinical sample (swab, aspirate or other)?
Thank you, we added the following: To achieve microbiological diagnosis, children were tested according to the decision of the evaluating physician. About SARS-CoV-2, all children in both centers received rapid antigenic SARS-CoV-2 testing, as for national anti-Covid policies. For the other respiratory viruses, we used ì the Allplex TM Respiratory Panel 1, 2 3 Seegene System is a molecular method for the genome detection and typing of the following Respiratory Viruses: Respiratory Syncytial Virus (RSV A and B), Influenza Virus ((Flu A and B), Parainfluenza Viruses (PIV 1, 2, 3 and 4), Adenovirus (AdV), Enterovirus (HEV), Metapneumovirus (MPV), Rhinovirus (HRV), Bocavirus (HBoV), Coronavirus (CoV NL63, 229E, OC43). Tests were dana on nasal swabs.
The following sentence should be deleted and only the parameters that showed a significant difference should be listed....“Although not always statistically significant, patients infected with SARS-CoV-2 showed a trend toward a higher incidence of ex-premature status, a higher SpO2, lower need for Oxygen support, and a tendency for lower admission rates to ward and PICU.“
Changed, thank you
Discussion:
„Interestingly, significantly less children had SARS-CoV-2 as an aetiologi-cal agent of bronchiolitis and children during the second season were significantly younger and without comorbidities.“ Please add the corresponding p values.
Thank you, we have added the following: “Having Covid as aetiological agent of bronchiolitis showed strong association with season (p<0.001) while having RSV as aetiological agent showed no correlation (p=0.103). Children in the second season were significantly younger (p=0.0094) and without comorbidities (p=0.001)”
Minor comments:
1) The number at the beginning of the sentence should be written as a word (i.e. nine hundred, eleven, thirty three etc...); numbers up to ten are also written as a word (i.e. two).
Thank you, changed
2) Accompanying the tables, abbreviations must be explained as well as when they appear for the first time in the text
Done, thank you